

# Simultaneous estimation of Ocean mesoscale and coherent internal tide Sea Surface Height signatures from the global Altimetry record

Clément Ubelmann[1], Loren Carrere[2], Chloé Durand[2], Gérald Dibarboure[3], Yannice Faugère[2], Maxime Ballarotta[2], Frédéric Briol[2], and Florent Lyard[4]

[1]Ocean-Next, 90 chemin du Moulin, 38660 La Terrasse, France
[2]Collecte Localisation Satellite, 11 rue Hermès, Parc Technologique du Canal 31520 Ramonville Saint-Agne France
[3]Centre National d'Etudes Spatiales, 18 avenue Edouard Belin, 31401 Toulouse, France
[4]LEGOS, 14 avenue Edouard Belin, 31400 Toulouse, France

**Correspondence:** Clément Ubelmann (clement.ubelmann@ocean-next.fr)

**Abstract.** This study proposes an approach to estimate the Ocean Sea Surface Height signature of coherent internal tides from 25 years of along-track altimetry record, with a single inversion over time, resolving both internal tide contributions and mesoscale eddy variability. The inversion is performed through reduced-order basis with conjugate gradient resolution. The particularity of this approach is to mitigate the potential aliasing effects between mesoscales and internal tide estimation from

the uneven altimetry sampling (observing the sum of these components) by accounting of their statistics simultaneously, while other methods generally use a prior for mesoscales. The four major tidal components are considered (M2,K1,S2,O1) over the period 1992-2017 on a global configuration. From the solution, we use altimetry data after 2017 for an independent validation, to evaluate the benefits of the simultaneous inversion, and also to compare the skills with an existing model.

## 1 Introduction

Ocean internal gravity waves forced by barotropic tidal currents, known as Internal Tides, have signatures on Sea Surface Height (SSH) at scales of 150km wavelength and below (e.g. Ray and Mitchum (1997)). Their vertical extension splits in several modes with specific propagations modulated by vertical stratification at first order. These waves, generated in phase with the barotropic tidal current, can remain phase-locked (called coherent hereafter) or phase modulated by seasonnal (Xinhua et al. (2020)) or mesoscale (Ponte and Klein (2015)) variability. If overall, 70% of the internal tide energy would be phase

modulated (Zaron (2017)) and challenging to resolve from the present along-track Altimetry record, the 30% coherent can be estimated from the series of more than 25 years. Several empirical approaches have been developed so far, from harmonic analysis (Ray and Zaron (2016)) to more sophisticated plane-wave fit (Zhao (2019), Zaron (2019)). Data assimilative Ocean General Circulation Models are also able to resolve internal Tides (Arbic et al. (2018)) accurately in amplitude, but their phase prediction seems not yet as accurate as the empirical approaches mentioned above, as reported in Carrere et al. (2020).

For the sake of improving the empirical approaches , we propose to tackle in this study the issue of mesoscale aliasing in the internal tide estimations. Indeed, the mesoscale signal is overall an order of magnitude above internal tides (these later being as small as 2cm of typical amplitude), potentially introducing some aliasing effects when observed by a constellation





of a nadir satellites. To mitigate this issue, most of approaches use separate estimates of mesoscales such as the DUACS maps distributed by AVISO and CMEMS (Taburet et al. (2019)) to substract to the altimetry data used for internal tide analysis.

However, it is known that these mesoscale maps may also be contaminated themselves by internal tide aliasing (Zaron and Ray (2018)) potentially affecting the final estimations. In this present study, we propose a simultaneous estimation of mesoscales and internal tides to optimize the mutual aliasing issues. It is based on a single massive inversion over large temporal windows, achieved thanks to a mode decomposition in reduced rank.

In the first section, we present the principle of the simultaneous estimation, with illustrations from idealized one-dimensional

synthetic signals. Then, in section 2, the application to Altimetry data through the mode decomposition is detailed, followed by the results and validation with independent data. Finally, we will discuss the perspectives, in particular to handle non-stationary internal tides with future wide-swath Altimetry data.

## 2   The simultaneous estimation : principle and illustrations

The estimation of pure harmonics from partial observations with errors or additional signals is generally performed with

harmonic analysis. However, this inversion does not account for particular covariances of the errors or additional signals. When these later are white random noise, the impact is minimized and the harmonic analysis remains the most optimal way. However, when we have the knowledge of covariances for the additional signals, especially if they are non-zero between observation coordinates, the estimation can be improved with a full consideration of the signals in the inversion.

### 2.1   General principle

Here, we present the general inversion formula for $N$ independent signals of different nature, whom the sum is observed.

We assume $N$ state vectors to estimate, $\mathbf{x_k}, k = 1..N$, and partial observations $\mathbf{y}$ of the sum of the components, that can be written as follows:

$$\mathbf{y} = \sum_{\mathbf{k=1}}^{\mathbf{N}} \mathbf{H_k x_k} + \epsilon \tag{1}$$

where $\mathbf{H_k}$ are the observation linear operators and $\epsilon$ is an observation error.

If we note $\mathbf{H} = [\mathbf{H_1}, .., \mathbf{H_N}]$ and $\mathbf{x} = [\mathbf{x_1^T}, .., \mathbf{x_N^T}]^\mathbf{T}$, then the observation vector writes: $\mathbf{y} = \mathbf{Hx} + \epsilon$ and the application of the OI formula gives:

$$\mathbf{x^a} = \mathbf{BH^T}(\mathbf{HBH^T} + \mathbf{R})^{-1}\mathbf{y} \tag{2}$$

where $\mathbf{B}$ is the covariance matrix of $\mathbf{x}$.

If we assume no cross-covariances between the two components, the matrix $\mathbf{B}$ writes:





$$\mathbf{B} = \begin{bmatrix} \mathbf{B_1} & \dots & 0 \\ \vdots & \ddots & \vdots \\ 0 & \dots & \mathbf{B_N} \end{bmatrix} \quad (3)$$

where the $\mathbf{B_k}$ are the covariances for $\mathbf{x_k}$.

## 2.2 Illustrations

The formulation presented above can be easily tested on a one-dimensional synthetic case, in particular to evaluate the gain of
the simultaneous inversion as presented, rather than separate inversions. Let's consider a one-dimensional signal, constituted
by the sum of a broadband component following the spectrum shown in black on the top panel of Figure 1 (whose integral
equals 1), and a pure harmonic of amplitude 1 as represented in gray on the same panel. A signal with these characteristics can
be randomly generated following the method described in Ubelmann et al. (2014), Annex 1, assuming random and independent
phases between Fourier harmonics. One example is represented on the middle panel of Fig. 1, by the thick black curve for the
broadband component and by the thin gray curve for the sum of the two components.

Now let's assume partial observations of this total signal, as represented by the black dots on the figure, here at random
times, with typical occurrences of 2-3 every 10 days and random observation errors of 0.01.

From these observations, we will illustrate the simultaneous inversion method aaplied to the 2 components (N=2 in Eq. 1 to
3), compared to separate estimations.

For the broadband part, $k = 1$, the state vector $\mathbf{x_1}$ is the broadband signal on the 1D-grid. $\mathbf{H_1}$ is a linear interpolator of $\mathbf{x_1}$
on the observation coordinates. The covariance matrix $\mathbf{B_1}$ is written in the grid space, with values following the covariance
function given by the inverse Fourier transform of the signal spectrum. It is therefore in this example perfectly optimal.

For the narrow-band part, $k = 2$, since we search here for a single harmonic, the state vector $\mathbf{x_2}$ can be reduced to vector of
2 elements, representing the amplitudes of a sine and cosine respectively. If $\mathbf{t_o}$ is the vector of observation time coordinates,
the observation operator $\mathbf{H_2}$ writes:

$$\mathbf{H_2} = \begin{bmatrix} cos(\omega \mathbf{t_o}), & sin(\omega \mathbf{t_o}) \end{bmatrix} \quad (4)$$

and the covariance matrix $\mathbf{B_2}$ of size 2 by 2 for $\mathbf{x_2}$ writes:

$$\mathbf{B_2} = \begin{bmatrix} \sigma_2^2 & 0 \\ 0 & \sigma_2^2 \end{bmatrix} \quad (5)$$

where $\sigma_2$ is the expected variance of the harmonic signal.

We used formula 2 to estimate simultaneously $\mathbf{x} = [\mathbf{x_1}, \mathbf{x_2}]$ from the vector of observations $\mathbf{y}$, with a diagonal $\mathbf{R}$ matrix
consistent with the 0.01 STD random error added to the observations. The inversion is made globally (no localization) since
the harmonic part has non-local covariances.





**Figure 1.** Top panel: power spectral density of the ground-truth one-dimensional signal, consituted by a low frequency continuum (black line) and by a high-frequency peak representing a single harmonic of unitary amplitude. Middle panel: time series of the ground-truth (generated randomly following the top panel spectra) in gray, with the low-frequency component in black line, and its three estimations (in colors as indicated in the legend) from the synthetic observations represented by the black dots. Bottom panel: time series of the high-frequency ground-truth with its three estimations (in colors as indicated in the legend).

In parallel, we run the so called 'separate estimations' where the inversions are applied separately for $x_1$ and $x_2$ (with $B_1$ and $B_2$ covariance matrices respectively) and an $R$ matrix accounting for observation error plus the representativity error of the unconsidered component. For the harmonic part, this separate estimation is quite similar to a harmonic analysis.



We also run the so-called 'sequential estimations', consisting in separate estimations with a modified vector of observation, whom the separate estimation of the unconsidered component has been removed. Note that this sequential estimation is similar to what is generally done for internal-tide estimations with previous removal of mesoscale signal (e.g. CMEMS/Aviso mesoscale maps).

The results of the different solutions are shown on Figure 1, in blue for the separate estimations, in green for the sequential
estimations and in red for the simultaneous estimations. Qualitatively, the separate estimations clearly suffer from aliasing, for instance the blue curve on the middle panel (broadband estimation) is skewed toward observations affected by a given phase of the harmonic signal at the time of observation (high to low frequency aliasing). Similarly, the harmonic estimation is affected by the broadband signal values at the time of observations (low to high frequency aliasing). The separate estimation (green) mitigates these issues, but the simultaneous estimation (red) seems clearly the best. It is particularly striking for the broadband
estimation. A quantitative assessment has been done after repeating this experiment 100 times, with results presented on Figure 2. The residual estimation errors for the broadband signal, represented in the frequency domain , clearly confirm the qualitative assessment. For the harmonic estimation, the results are also confirmed (right panel) with an additionnal interesting point: if the independant estimation is the worse (more dispersion for the blue dots), it seems unbiased, while the sequential estimation, better than the separate at least in term of phase error, is clearly biased toward lower amplitudes. This can be explained because
in the sequential estimation of the harmonic, the first step (separate estimation of the broadband) was skewed toward the harmonic at observation locations. So the substraction of this broadband estimation tends to remove some signal in phase with the harmonic. This is precisely what we think happens to the estimation of internal tides from Altimetry data when mesoscale Aviso maps are subtracted to the data, mitigating aliasing but also introducing systematic under-estimations of internal tides. This illustration is of course pushed to a much higher extent, but certainly shows the potential weakness of present internal tide
estimations and suggests to implement a simultaneous estimation.

## 3   Application to the global Altimetry Record

For the case of coherent internal tides, the waves span over 30 years of satellite Altimetry, so a single inversion over time is not as trivial as in the illustration presented above. Even with suited spatial localization, the number of observations would exceed the limit of matrix invisibility (and even matrix storage). To overcome this issue, one can define covariances through
the construction of a reduced basis as presented in Ubelmann et al. (2021) and invert the system with conjugate gradients bypassing the explicit massive inversion.

### 3.1   The inversion in reduced space

This paragraph is a summary of the method presented in details in Ubelmann et al. (2021). For all component $k$, we consider $\Gamma_k$ the reduced basis of elements in the time-space domain to approximate $\mathbf{x_k}$ as follows:

$$\mathbf{x_k} = \mathbf{\Gamma_k}\eta_{\mathbf{k}} \tag{6}$$




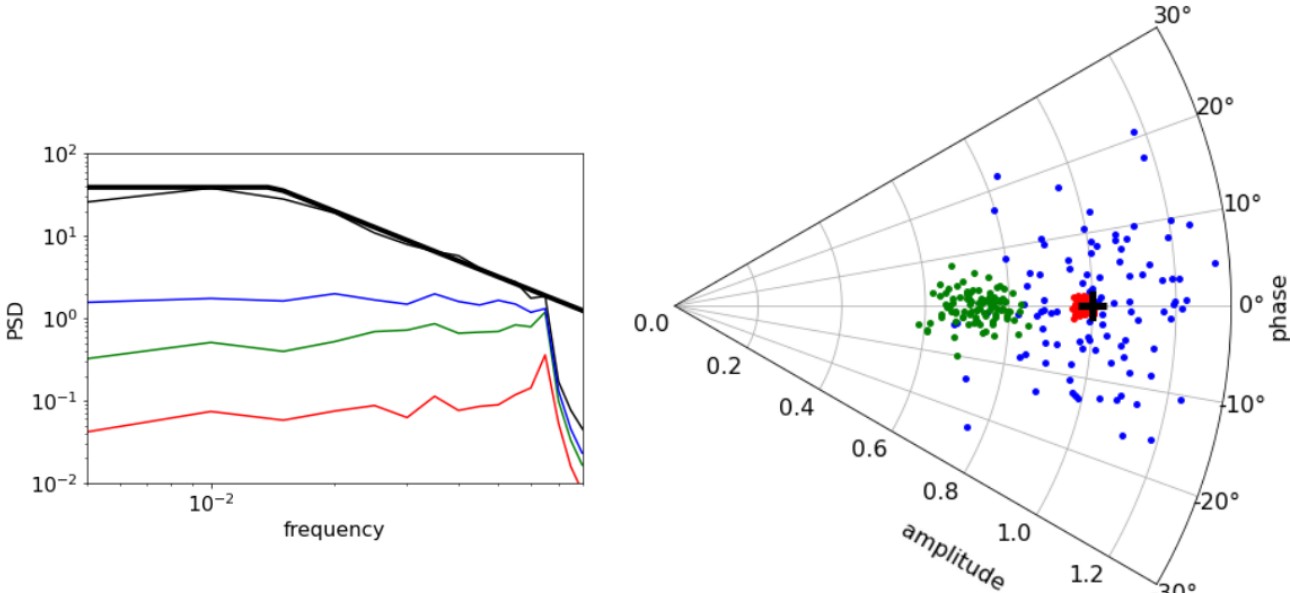

**Figure 2.** Left: Mean (averaged over the 100 realizations) power spectral densities of the LFC ground truth (thin black) and the LFC estimation error (estimation minus ground truth) for all three estimations (independant, sequential and simultaneous in blue, green, red respectively). The thick black curve represents the theoretical LFC ground-truth spectrum. Right: Amplitude and phase (in the complex domain) of the FHP harmonic, represented by the black cross for the ground truth, and by the green, blue and red points for the 100 realizations of the independant, sequential and simultaneous estimations respectively.

where the $\eta_{\mathbf{k}}$ are the state vectors in parameter space

Then, if we note $\mathbf{G_k} = \mathbf{H_k}\mathbf{\Gamma_k}$ where $\mathbf{H_k}$ is the observation operator from grid to observation space (tri-linear interpolators), the observation vector $\mathbf{y}$ writes:

$$\mathbf{y} = \sum_{\mathbf{k=1}}^{\mathbf{N}} \mathbf{G_k}\eta_{\mathbf{k}} + \epsilon = \mathbf{G}\eta \tag{7}$$

where $\mathbf{G} = [\mathbf{G_0}, \ldots, \mathbf{G_N}]$ and $\eta = [\eta_{\mathbf{0}}, \ldots, \eta_{\mathbf{p}}]$. The application of the OI formula to 7 gives:

$$\eta^{\mathbf{a}} = \mathbf{Q}\mathbf{G}^{\mathbf{T}}(\mathbf{G}\mathbf{Q}\mathbf{G}^{\mathbf{T}} + \mathbf{R})^{-\mathbf{1}}\mathbf{y} \tag{8}$$

where $\mathbf{Q}$ is the covariance matrix of the state vector $\eta$. Since no covariances are assumed between components, $\mathbf{Q}$ has the form:





$$\mathbf{Q} = \begin{bmatrix} \mathbf{Q_1} & \dots & 0 \\ \vdots & \ddots & \vdots \\ 0 & \dots & \mathbf{Q_N} \end{bmatrix} \tag{9}$$

where the $\mathbf{Q_k}$ will be assumed diagonal.

The equivalent covariance models are:

$$\mathbf{B_k} = \mathbf{\Gamma_k} \mathbf{Q_k} \mathbf{\Gamma_k^T} \tag{10}$$

These covariance models will rely on the structure of elements in $\mathbf{\Gamma}$ and the variances chosen in $\mathbf{Q}$ for each of the elements, as detailed in the following. The Mesoscales will constitute the first component, and the internal tides will be expressed as

several components according the the tidal frequencies and vertical modes.

### 3.1.1   The decomposition basis for Mesoscales

The basis for mesoscales is the same as presented in Ubelmann et al. (2021), where the analytical formula of the ensemble of basis elements are given. Here, we only provide an illustration for comparison with the Internal-Tide basis described in details in the next section. The left panel of Figure 3 shows one of these elements for a particular wavenumber (of 150km),

with a typical time extension of 10 days. The ensemble of components characteristics and the diagonal terms of the $Q$ are designed to approach equivalent covariances of Ocean mesoscale altimetry signals. This is also illustrated on the left panel of Figure 4, where the representer features standard shapes of Altimetry mapping covariance function with a negative lobe (Traon et al. (1998)). Note that the representer is not perfectly symmetric because geographical variations of the prescribed element variances ($Q$) introduce some in-homogeneity.

### 3.1.2   The decomposition basis for Internal Tides

The basis for Internal Tides presents some similarities in the construction, but accounting for specific aspects of the internal tide dynamics, in particular the forcing frequency and the horizontal dispersion relation.

For M2 and K1 forcing frequency, the contribution of the first two vertical modes are considered, because they have a major contribution on SSH at the horizontal scales where Altimetry observations are accurate (limited to 50-80km along-track,

Dufau et al. (2016)) and also higher modes would have certainly more complex interactions. We also assume no statistical dependence between the modes so they are considered as independent component addding to the parameter dimension with zeros off-diagonal terms in the $Q$ matrix. For S2 and O1, only mode-1 are considered since their signal is weaker and mode-2 seemed challenging to resolve accurately.

For mode-1 (of any forcing frequency), we built a decomposition of plane waves following the dispersion relation between

the time frequency $w$ and spatial wavenumber $k$:

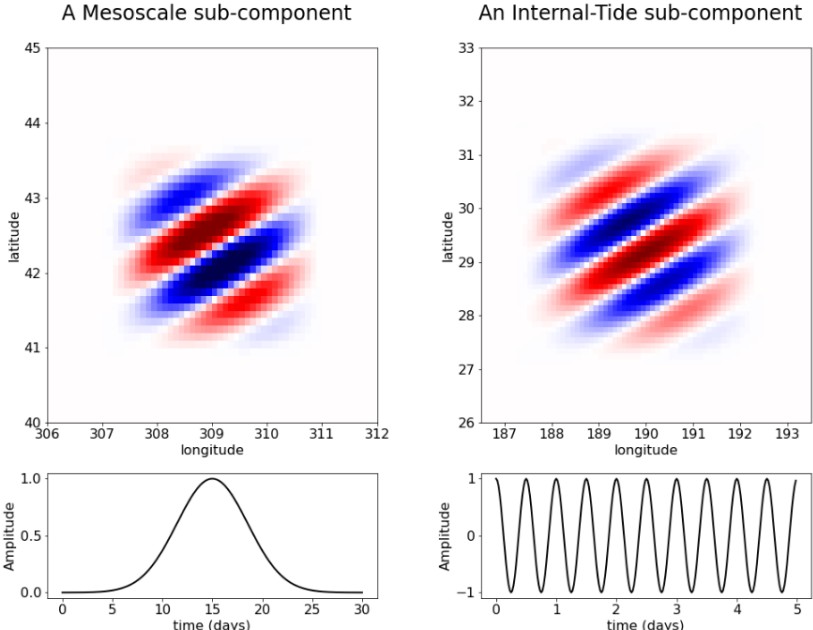

**Figure 3.** Illustration of one element of the basis for mesoscale broadband signal (left) and for internal-tide narrowband signal (right). The upper panels show the element in space. For mesoscales, the represented element has a dominent wavelength and direction, but the ensemble of elements spans over all wavelengths and directions. For internal-tides, the represented element has the dominent wavelength of the first baroclinic mode at the location considered (this is a mode-1 element) and a given direction, but the ensemble of elements spans also over both mode-1 and mode-2 wavelength, and all directions. The lower panels show the element in time. For mesoscale, the time extension is local as indicated by the Gaussian-like shape. For Internal-Tides, the element persists in time with a sine modulation according to internal tide dynamics.

$$w^2 = k^2 c^2 + f^2 \qquad (11)$$

from which we can derive the phase velocity $c_p$ (e.g. Zhao (2019)) :

$$c_p = \frac{w}{\sqrt{w^2 - f^2}} c \qquad (12)$$

where $f$ is the Coriolis frequency and $c$ the first baroclinic mode phase speed. We specified regional variations for $c$ following the Chelton et al. (1998) climatology, with values typically spanning between 1 m/s in shallow or weakly-stratified regions to 3.5 m/s in the tropics and mid-latitude Gyres. If lower values are obviously present in very shallow areas, these later are not defined in the climatology and we did not consider the presence of internal tides in those areas. The plane-waves are



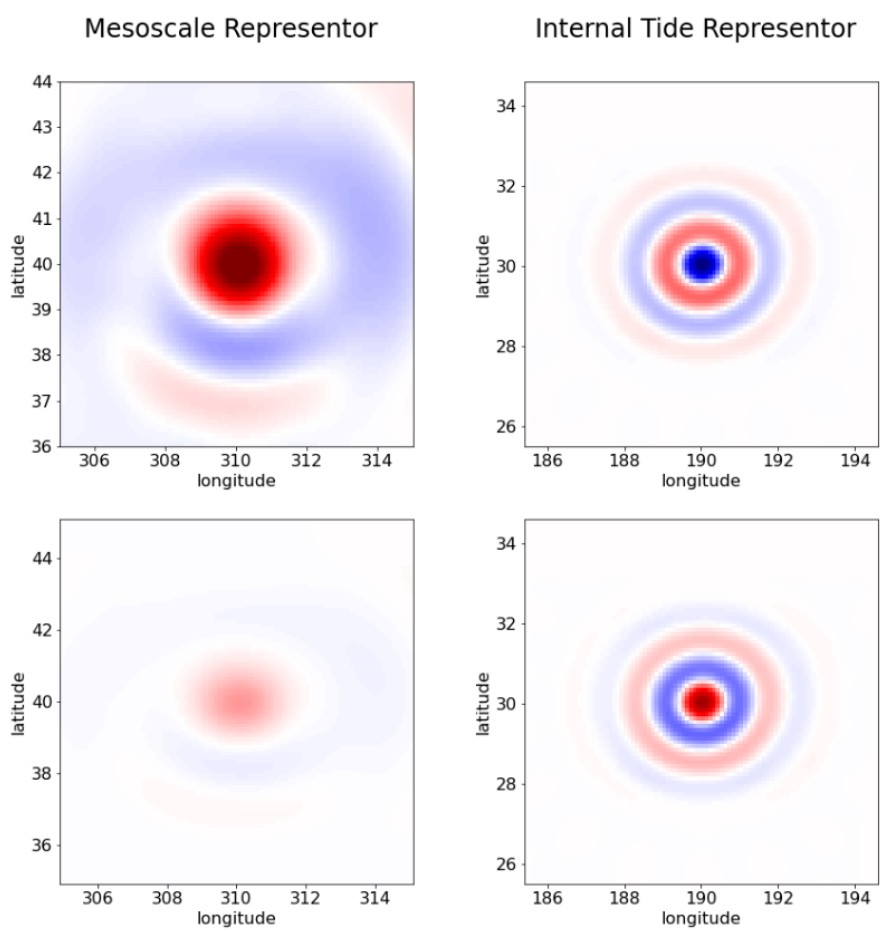

**Figure 4.** Illustration of the equivalent representer at a given location, for the mesoscale basis (left panels) and for the internal tide basis (right panels) as a function of space, for zero time lag (upper panels) and for 15-day time lag (lower panels)

defined as purely persistent in time, as illustrated on the bottom-right panel of Figure 3, at exactly the forcing frequency considered for each component. However, in space, they are localized with a Hamming window at 3 spatial wavelength (see
upper right panel), to account for geographical variations due to topography (sources and tinks), inaccurate phase speed and any unconsidered physical interactions. The whole domain is paved with similar elements, and 12 angles for the plane wave directions. The number of angles must be sufficient to cover all possibility but not to large to avoid redundancies; its optimal value is related to the Hamming window extension w.r.t. spatial wavelength. Finally, for each angle, two elements are defined (with $\pi/2$ shift) to allow the degree of freedom on the phase fit.
Using such a basis with a $Q$ diagonal matrix with constant terms (equi-probability for each element), the representer $\Gamma[i,:]Q\Gamma^T$ for a given point of the grid at coordinates (190,30) is shown on the right panels of figure 3, for zero time lag (ummper panel) and 15-day time lag (plus half the tidal period). The wavy pattern is clear on the representer shape, as opposed to





mesoscales, since the waves are narrow-band in space, by construction. There are also no attenuation in time (just phase fluctuations) as opposed to mesoscales, because the waves are infinite-narrow-band, also by construction.

For the vertical mode 2 (only M2 and K1 forcing frequencies considered), the basis are constructed exactly the same way, with a second vertical baroclinic phase speed $c_2 = c/2$. The phase speed from Eq. 12 is therefore divided by two resulting in spatial wavelength twice smaller as for the mode 1. The spatial Hamming windows are still scaled on 3 wavelengths. The total number of elements is therefore doubled in the meridional and zonal directions.

### 3.1.3  The inversion

Since the size of parameter space are be much smaller than the size of observation space, it is interesting to consider the Sherman-Woodbury transformationto Eq. 10 giving the equivalent following expression for $\eta^a$:

$$\eta^{\mathbf{a}} = (\mathbf{G^T R^{-1} G + Q^{-1}})^{-1} \mathbf{G^T R^{-1} y} \tag{13}$$

The problem is solved on 15° by 15° tiles separately, paving the whole globe with 2° overlaps to allow smooth transitions on the final concatenation of the solution.

On each tile, the first step consists in filling the $G$ matrix for mesoscales and internal tides following the basis decomposition described above, over the whole Altimetry record. It consists, for each element, in an interpolation of the element value at each observation point. In practice, this interpolation is computed directly with the analytical expression of the element. For mesoscales, the number of elements (matrix column) is extremely large (10**9 over 25 years and global domain) but the matrix is also extremely sparse. Indeed, as illustrated on the left panel of Figure 5, very few observation points are included in a given

element extension (local in time and space) which allow a manageable size. For internal tides, the number of elements is much smaller (10**7) as the elements are local in space only but persist in time. The matrix is therefore more dense (sparse in space only), as illustrated on the left panel of figure 5 : all observation points of the 25-years of altimetry are included the element. The global size of the matrix is also manageable.

The second step is the resolution of Eq. 13, as explained in details in Ubelmann et al. (2021), to converge toward the solution

after typically 100 iterations with a conjugate-gradient method. Then, the third step is the projection of the solution in the grid space, formally $\Gamma\eta$ but in practice this is computed sequentially by adding each column of $\Gamma$ (i.e. a given element) constructed online and multiplied by the $i^{th}$ component of $\eta$, bypassing the storage of $\Gamma$. The mesoscale solution is written at daily posting on a 1/8 degree grid in the zonal and meridional directions, while the internal tide solution is written on the same spatial grid but at a single date of reference (January 1st, 1990), at midnight and midnight plus half the tidal frequency of the mode. Since

the internal tide solution is coherent, these two fields are sufficient to reconstruct the solution at any time.

To overcome the large computation and storage of the $G$ matrix, and in particular because the iterative matrix multiplications are much more efficient if $G$ is stored in RAM, these three steps are performed on a supercomputer, occupying a total memory approaching 2 To RAM shared by 200 threads. The $G$ matrix is segmented, with appropriate communications at each computation of matrix products.





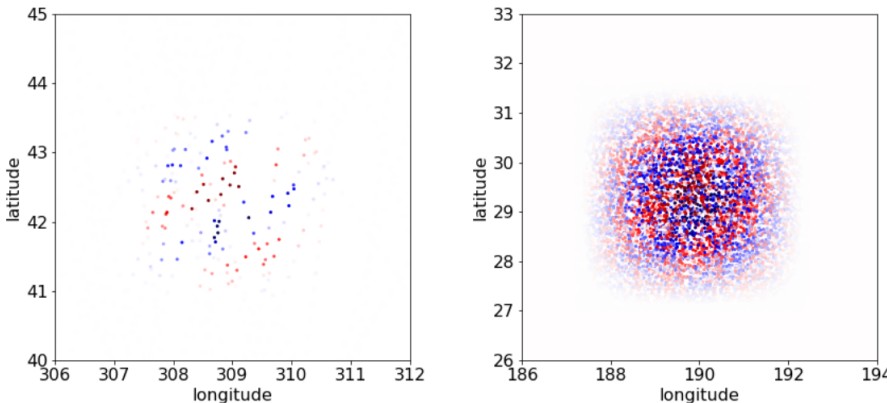

**Figure 5.** Illustration of the same elements as presented in Figure 3 (from Masoscale basis on the left and from Internal-tide basis on the right), visualized in the observation space. The coloured dots represent the non-zero values of a column of the $G$ matrix.

Finally, from the solutions on the overlapping 15° by 15° boxes, a continuous global solution is computed by linearly interpolating the solution in the overlapping zones, with a weight ratio proportional to the boundary relative distances.

## 3.2   Results and validation

Both mesoscale and coherent internal tides estimations are provided from the Altimetry inversion. The mesoscale solution, supposed to be less affected by internal-tide aliasing, is not analysed in this paper focused on the internal-tide solution. An 
illustration on figure 6 shows the two mapped fields in the North Atlantic. The characteristic of each signal are very clear, with large mesoscale eddies on the top, dominating the signal variance, and the internal tides at the bottom, with clear generation sites showing up at expected locations.

As mentioned above, the internal tides are resolved simultaneously, but separately for each components (tidal frequencies and vertical modes). The illustration of all these components is particularly interesting in some regions where several components 
dominates such as the Philippine Sea as shown on figure 7. As expected, the wavelengths for mode-2 are about twice smaller than mode-1, and the wavelengths for K1 are twice longer than M2 since the time period is doubled. Note the K1 components dominates few areas, and supposedly in the 30S-30N tropical band. After verifying that no significant signal projected on the K1 frequency beyond 30N and 30S, we run the final inversions without specifying the K1 components beyond 30N and 30S.

In the following, we use the internal tide fields (written at the reference date) in a prediction mode assuming the stationary 
persists over the period September 2017 - January 2019 where we use independent Altimetry data. To assess the quality of the reconstruction, a classical metric is the variance reduction of the along-track Sea Level Anomalies before and after applying the prediction (Carrere et al. (2020)). This later can be computed either over boxes or smaller pixels.



**Figure 6.** Snapshots of the reconstructed solutions on May 20, 2017 at 00 UTC in the North Atlantic basin, for the mesoscale component (top panel) and for the internal tide component (lower panel).

### 3.2.1  The benefits of simultaneous estimations

Before analysing the global solution, we propose to look at dedicated experiments where the estimation of internal tides
are not simultaneous with mesoscales (only internal tide components are considered in the **G** matrix), a first experiment ignoring mesoscales, and a second considering mesoscales, but as a prior inverted separately. The results are shown on Figures


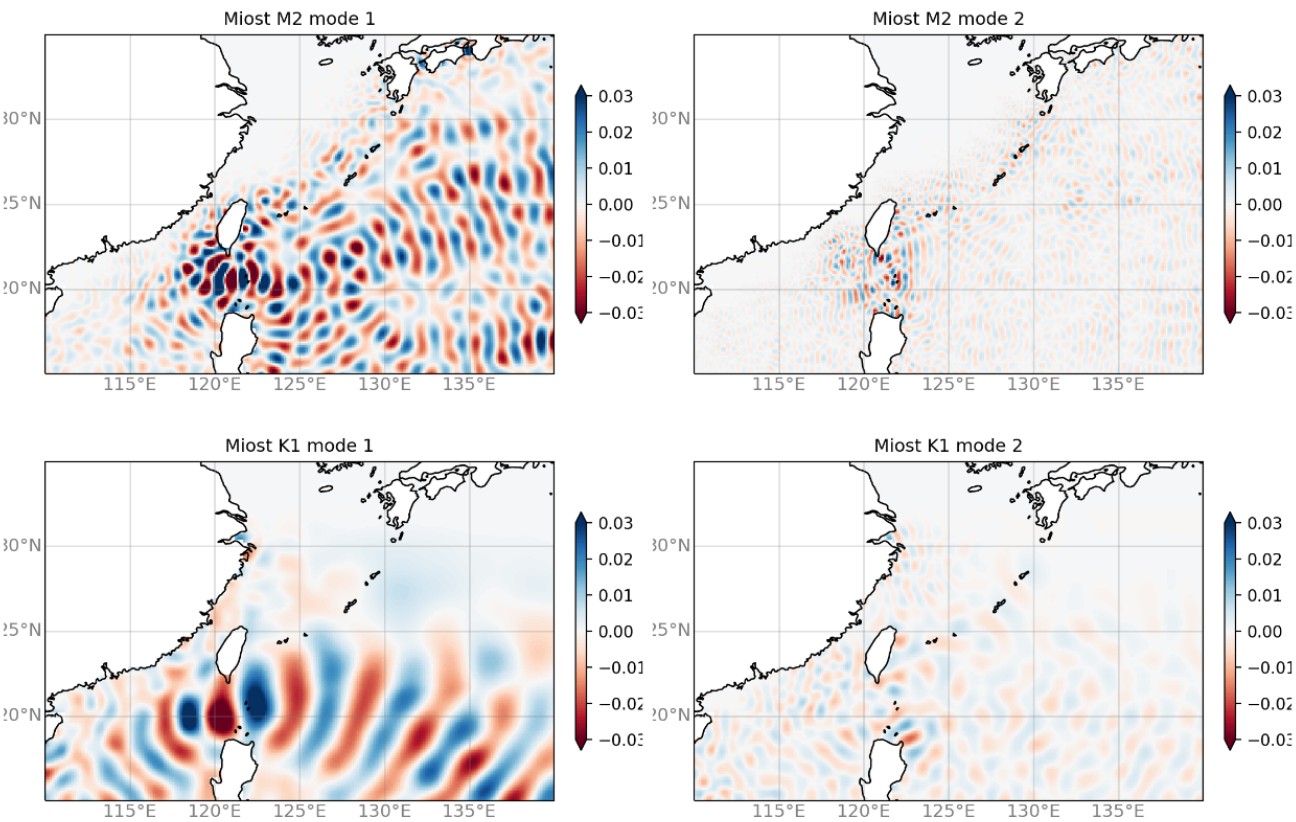

**Figure 7.** Snapshots of the reconstructed internal tide solutions in the Philippine Sea for M2 (top panels) and K1 (bottom panels), for the first mode contribution (left) and the second mode contribution (right).

8 and 9, the two experiments are labelled 'MIOST-IT' and 'MIOST-MS then MIOST-IT', compared to the simultaneous experiment 'MIOST-MSIT'. The Figure 8 is in the Hawaii region where mesoscale contamination is supposed to be moderate, for quite intense coherent internal tides. The effect of the simultaneous inversion is notable with a reduced variance (green

bars) exceeding the first two experiments. As expected, the variance of the solution (blue bars) is higher in the first experiment (mesoscale contamination) while in the second experiment, the variance is much lower, probably because the prior mesoscale field contained aliased internal tides therefore reducing the apparent internal tides. The Figure 8 is in the Gulf-Stream region with intense mesoscales and potentially low coherent internal tides. Here, the effect of simultaneous inversion is even stronger. In absence of mesoscale consideration, some internal tides are estimated in the Gulf-Stream, but the reduction variance is

negative, meaning that the estimations feature more errors than actual signal. This is also clearly the effect of mesoscale aliasing. Note the importance of using independent data over an independent period out of the analysis window : if we used the data during the analysis, the variance reduction would be positive by construction. In the second experiment, the same effect as in the Hawaii case is observed, and amplified : the internal tides are very attenuated because the prior mesoscale estimates

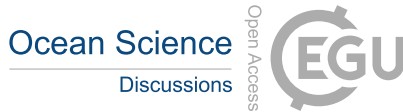

are contaminated in phase with the internal tides during the period of analysis. Finally, the simultaneous inversion seems to do

quite a good job in the Gulf Stream, resolving some internal tides in the continental slope area, although probably attenuated

since the variance reduction exceeds the signal variance (the $Q$ matrices for mesoscales and Internal-tides may be reajusted to

tolerate a bit more internal tides).

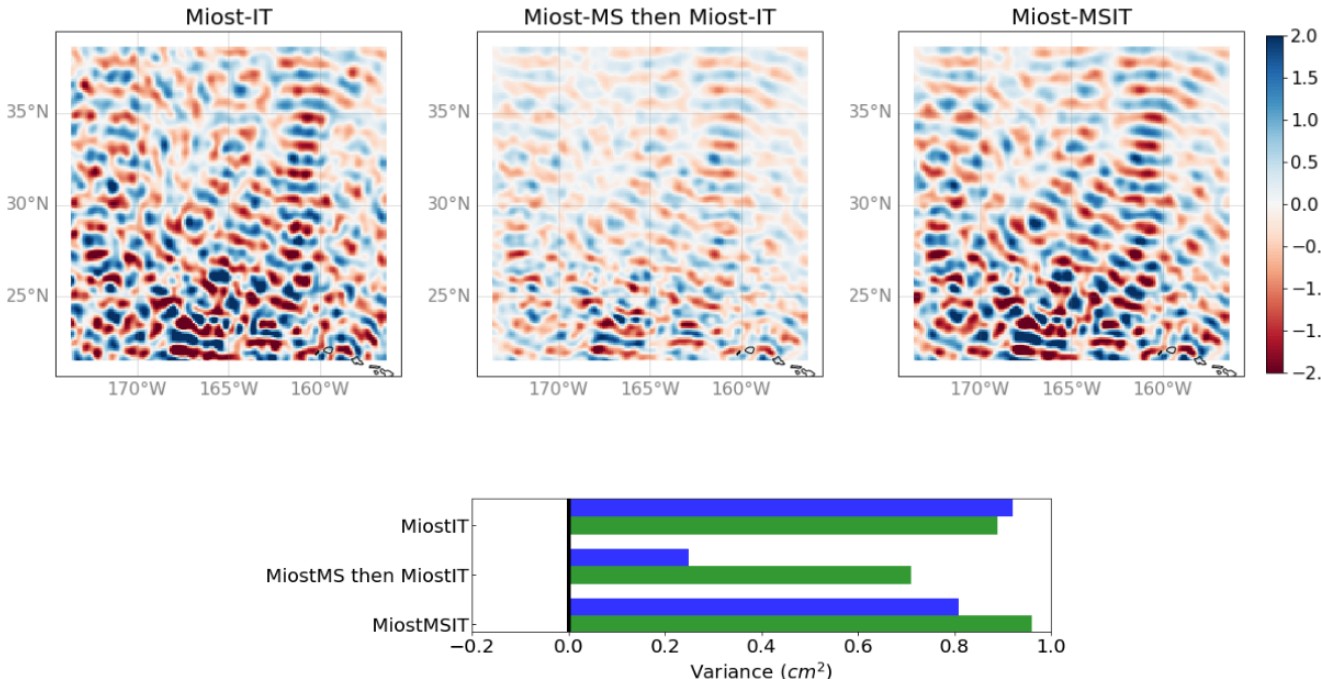

**Figure 8.** Upper panel: snapshots of the M2 internal tide field north of the Hawaii ridge for three different experiment : in the first one ('MIOST-IT, 'left panel) only the internal tide components have been considered in the inversion. In the second one (MiostMS then MiostIT, middle panel) the internal tide mode inversion has been applied on observations whom the field from the mesoscale inversion has been previously removed. In the third one (MIOST-MSIT, right panel), the simultaneous inversion has been applied. Lower panel: signal variance (in blue) and explained variance after applying the signal as a correction to independent altimetry data (in green) for the three experiments mentioned above.

### 3.2.2 Global validation

The main simultaneous-inversion experiment, that featured the expected mitigation of aliasing issues over independent or

sequential estimations, is now validated globally and compared with an existing internal tide solution from Zaron (2019).

The global SLA variance reduction has been computed over 2° by 2° cells over the September 2017 - December 2018 multi-satellite Altimetry data, a period during which none of the estimations have seen the data. The results are shown on Figure 10, for the two major components M2 and K1 (including 1st and second baroclinic modes). First, there are no regions (except a few



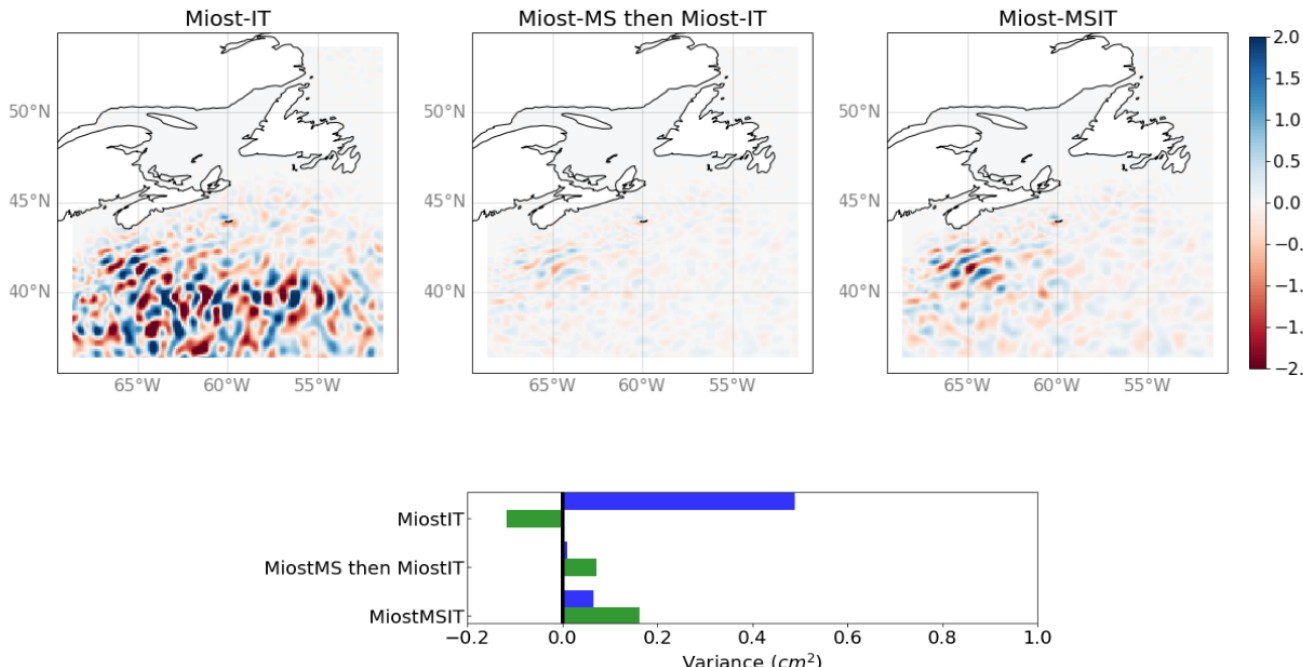

**Figure 9.** Same as Figure 8 in the Gulf Stream area

red pixels) where the estimation fails : the variance reduction is everywhere either positive (blue meaning negative difference,

i.e. positive reduction) or absent. It is very positive in the expected sites of internal-tide generation where internal tides are supposedly coherent. The Equatorial regions, known to feature phase-varying internal tides because of strong current, have indeed no significant variance reduction (no signal neither, not shown). The comparison with the Zaron (2019) model is at first order very similar, which we consider as a success since the approach was new and could have led to unexpected issues. Then, a closer look at the differences between the two estimates (lower panels) would suggest that the variance reduction is, on global

average, slightly higher in this study, especially in the regions of high internal-tide energy. Also, it might be of interest to note that the wave train resolved north of the Gulf Stream (see upper-right panel of Figure 8) shows up with a few blue pixels on the bottom-left panel of Figure 10, whereas the Zaron model does not provide signal here (flagged region). Finally, there are still some areas where the Zaron (2019) features more variance reduction, for instance with K1 in the Eastern Tropical Pacific.

The integrated variance reductions computed for Cryosat-2 mission have been integrated over the main regions of intense

internal-tide, and for M2 and K1 components available in both models, as shown on the Table 1. The reductions are indeed very close, and would confirm a slight advantage to the simultaneous inversion presented in this study.

**Figure 10.** Difference of Sea Level Anomaly variance between along-track data before and after applying the Internal Tide prediction, averaged over the period September 2017 - December 2018. Upper panels : with the simultaneous MIOST reconstruction, for M2 component (left) and for K1 component (right). Middle panels : same with the Zaron (2019) prediction model. Lower panels: difference between upper panels and middle panels.

## 4   Conclusions

The simultaneous inversion of mesoscale variability and internal tide signatures on Sea Surface Height has been successfully implemented globally, constituting in particular an additional empirical internal-tide model to the existing ones. It was designed





**Along-track SLA**

| Modèles comparés | Global | Tahiti | Hawaii | Madagascar | Gulf of Guinea | Luzon | NATL | NPAC |
|---|---|---|---|---|---|---|---|---|
| MIOST − ZERO | - 0.31cm² | - 0.81cm² | - 0.72cm² | - 0.83cm² | - 0.09cm² | - 1.71cm² | - 0.19cm² | - 0.36cm² |
| % | - 0.28% | - 2.02% | - 0.92% | - 0.79% | - 0.29% | - 0.94% | - 0.26% | - 0.64% |
| MIOST − ZARON_2019 | - 0.13cm² | - 0.10cm² | - 0.07cm² | - 0.22cm² | - 0.00cm² | - 0.29cm² | - 0.04cm² | - 0.03cm² |
| % | - 0.15% | - 0.25% | - 0.09% | - 0.21% | - 0.01% | - 0.18% | - 0.06% | - 0.05% |

**SSH Crossovers**

| Modèles comparés | Global | Tahiti | Hawaii | Madagascar | Gulf of Guinea | Luzon | NATL | NPAC |
|---|---|---|---|---|---|---|---|---|
| MIOST − ZERO | - 1.26cm² | - 2.19cm² | - 1.93cm² | - 2.20cm² | - | - 4.26cm² | - 0.24cm² | - 0.60cm² |
| % | - 2.40% | - 9.07% | - 7.13% | - 3.86% | - | - 5.00% | - 0.91% | - 2.74% |
| MIOST − ZARON_2019 | - 0.16cm² | - 0.43cm² | - 0.16cm² | 0.04cm² | - | - 1.12cm² | - 0.10cm² | - 0.02cm² |
| % | - 0.39% | - 1.92% | - 0.63% | 0.07% | - | - 1.42% | - 0.38% | - 0.09% |

**Along-track SLA**

| Modèles comparés | Global | Tahiti | Hawaii | Madagascar | Gulf of Guinea | Luzon | NATL | NPAC |
|---|---|---|---|---|---|---|---|---|
| MIOST − ZERO | - 0.06cm² | - 0.04cm² | - 0.04cm² | - 0.07cm² | - 0.01cm² | - 1.45cm² | 0.00cm² | - 0.01cm² |
| % | - 0.05% | - 0.10% | - 0.05% | - 0.07% | - 0.02% | - 0.79% | 0.01% | - 0.02% |
| MIOST − ZARON_2019 | - 0.05cm² | - 0.00cm² | - 0.03cm² | - 0.03cm² | - 0.02cm² | - 0.62cm² | 0.02cm² | - 0.02cm² |
| % | - 0.06% | - 0.01% | - 0.04% | - 0.03% | - 0.06% | - 0.39% | 0.05% | - 0.03% |

**SSH Crossovers**

| Modèles comparés | Global | Tahiti | Hawaii | Madagascar | Gulf of Guinea | Luzon | NATL | NPAC |
|---|---|---|---|---|---|---|---|---|
| MIOST − ZERO | - 0.46cm² | - 0.29cm² | - 0.08cm² | - 0.16cm² | - | - 1.04cm² | 0.00cm² | - 0.03cm² |
| % | - 0.88% | - 1.20% | - 0.30% | - 0.28% | - | - 1.22% | 0.02% | - 0.14% |
| MIOST − ZARON_2019 | - 0.02cm² | 0.01cm² | - 0.04cm² | - 0.13cm² | - | 0.52cm² | 0.15cm² | - 0.07cm² |
| % | - 0.05% | 0.04% | - 0.13% | - 0.23% | - | 0.67% | 0.93% | - 0.35% |

**Table 1.** Tables summerizing the variance differences when using independant Cryosat data, averaged over different regions of interest for Internal-Tides, if using MIOST solution only (comparison with no IT correction) and differences between the use of MIOST and the Zaron (2019) models. First panel : variance difference of the full SLA signal, when correcting from the M2 component. Second panel: variance difference of crossovers (ascendant/descendent) differences, still correcting from the M2 component. Third and fourth panels : same for K1 component.

to minimize some aliasing effects, by accounting for the statistics of mesoscales and internal tides in a global linear analysis framework.

The computation of uncertainties could be interesting next step. Indeed, it would be possible to provide estimations of errors, with respect to the covariance model prescribed through the mode decompositions for mesoscales and internal-tides (first given in the parameter space, but then projectable in physical space). This would be a future improvement, considering

that the successful quantitative validation with independent data was the most important milestone for this new internal-tide model.

The methodology applied in this study also opens the door for resolving uncoherent internal tides with future wide-swath Altimetry. This is obviously a challenge that will not be met everywhere, in particular where mesoscale fields dominate. However, we expect that in some regions, the implementation of new internal-tide components allowing phase fluctuations

could be constrained with high-resolution Altimetry data. Indeed, since the components have a temporal and spatial extension (making use of the full spatio-temporal stuctures of internal tides in a quite reduced space of low degrees of freedom), we hope that the 2D aspects of future data will be extremely useful.

*Data availability.* The internal tide solution from the MIOST analysis described in this article is available on the Aviso website at : https: //www.aviso.altimetry.fr/en/data/products/auxiliary-products/internal-tide-miost.html. A full description on how to use the data, in particular

on how to reconstruct the internal tide solution from frequency to temporal space is provided here : https://www.aviso.altimetry.fr/fileadmin/ documents/data/tools/hdbk_MIOST-IT.pdf

*Author contributions.* CU designed the methodology, coded the numerical implementation and wrote sections 1, 2 and 3.1, 3.2.1. LC provided expertise on internal tide during the methodology design, supervised the global validation (section 3.2.2), managed the data availability and wrote the data user handbook. CD run the global validation. GD provided altimetry data expertise and critical advice on the methodol-

ogy. YF helped with data managment, MB and FB provided support during computational implementation. Finally, FL provided support on theoretical aspects of internal tides.

*Competing interests.* The authors declare that no competing interests are present.

*Acknowledgements.* This study was funded by the Centre National d'Etudes Spatiales (CNES) as part as the DUACS project



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
