# Peer review of "Simultaneous estimation of Ocean mesoscale and coherent internal tide Sea Surface Height signatures from the global Altimetry record"

_Ocean Science, 2021_

## Author Comment (AC1)

We thank Edward Zaron for his very detailed and constructive review, with in-depth comments and suggestions very helpful to propose a second version of the manuscript. We propose here a point-by-point response, supported with new tests, computations and figures when necessary.

- *No details seem to be provided about the Q matrices for both the mesoscale and tidal signals. Can you provide, say, a pseudo-spectrum of variance that shows how Q varies as a function of wavenumber or horizontal mode number; or do you feel that this is represented equivalently with the physical-space representers shown in Fig 5? Can you provide maps of the mesoscale and tidal variance (i.e., the diagonal elements of the Q matrix)? Did you perform any tuning to adjust the ratio of Q and R, or the ratio of the tidal Q and the mesoscale Q?*

Indeed, the discussion about the Q matrix was not detailed enough in the manuscript, only mentioning that the diagonal terms of Q are designed to approach equivalent covariances of Ocean mesoscale altimetry signal. We propose a more detailed description, which is based on an input dataset of power spectral densities measured by the AltiKa along-track SSH anomalies. For a given dominant wavelength of the wavelet of the mesoscale decomposition, the corresponding column of the Q matrix has the energy of the integrated spectrum between half the preceding wavelength and half the following wavelength. The database of power spectral densities is illustrated on Figure 1 (in space for a given wavelength) and Figure 2 (in wavelength at a given location) of this response document.

[Figure]

**Figure 1 : power spectral density (in cm²/cy/km) at 500km wavelength from the along-track sea level anomalies of AltiKa satellite**

[Figure]

**Figure 2 : power spectral density (in cm²/cy/km) at 60°W, 35°N from the along-track sea level anomalies of AltiKa satellite**

As shown on the figures, there is an important spatial variability of the energy, and an energy decrease with the wavenumber (red spectrum). Note that the Altimeter noise was removed following Fu and Xu, 2012. There variabilities are therefore taken into account in Q, which allow a representer function (Figure 4 of the manuscript).

As for internal tides, in the Q matrix block, we did not consider any a priori on regional variability. We set up the diagonals of Q at 2cm squared for mode 1, 1cm squared for mode 2, for all tidal components, as we propose to explain in the new version of the manuscript.

*If I understand correctly, the estimate you compute using eqn (8) or (13) is biased towards zero. A symptom of this bias is the observation that the explained variance is larger than the signal variance (as you noted with regard to the bottom panel of Fig 8). Can you plot a map of this variance ratio and interpret it with regard to either the bias or the tuning of Q? Would you consider using a non-zero estimate of the tide or the mesoscale in order to reduce this bias?*

Yes, this is a good point,  we plotted below (Figure 3) the variance of the signal (top panel) and the difference between this latter and the variance reduction after applying the correction (we found that plotting the difference was more explicit than the ratio, but the message is the same) : indeed,  the bottom panel has more blue color than red, meaning that the variance reduction is stronger than the variance of the signal itself. However, some clear red spots are present, in particular in the Kurushio. One hypothesis is that the mesoscale leakage (although mitigated) gets sometimes higher that the bias toward zero.

[Figure]

[Figure]

**Figure 3 : Signal variance (cm\*\*2) of the M2 estimation binned on a 2° by 2° grid (top panel) and difference between the signal variance and the reduction of variance after applying the signal to the independent data (lower panel). The blue zones indicate a higher variance reduction than the variance of the signal itself.**

By curiosity, we did the same figure, but with the HRET model (Figure 4). We still note the presence of red spots, and some of the blue spots (variance reduction higher than the signal variance) are more blue than in MIOST, meaning that the bias feature is present in the two models.

[Figure]

[Figure]

**Figure 4 : Same as Figgure 3, for the HRET (Zaron) M2 signal.**

It is also interesting to plot the difference of signal variance between MIOST and HRET (lower panel of Figure 5) compared with the difference of variance reduction. We can note that the signal in MIOST is significantly stronger (much more red than blue in the upper panel) but the variance reduction is only slightly stronger (lower panel). This would suggest that MIOST brings overall more signal, but also more error with a light net advantage in variance reductions. This also supports that the two models have probably their own advantages.

As for using a non-zero prior estimate of the tides or mesoscale, we are not sure this would mitigate the bias issue since those estimate have to be generated with Altimetry observation containing both signals anyways, the the bias issues would be still there. We would be happy to discuss more this question is this is not clear.

[Figure]

[Figure]

**Figure 5 : difference of signal variance between HRET and MIOST (top). The red zones indicate where the MIOST signal has more variance. Difference of variance reduction (as the bottom-left panel of Figure 10 in the paper)**

*No details are provided with regard to the time-dependence of the tides, except for equation (4). It appears that the nodal modulations have been omitted, but this is a substantial effect over a 25 year record. Properly accounting for this would probably further increase the explained*

*variance of the tidal estimates with respect to both the validation data and the assimilated data; and it should furthermore reduce the (low) bias of the tidal amplitudes.*

This is correct, and for Eq.4, we rigorously assume stationarity in the tidal forcing and solution we seek as this is an idealized test.
However, in the main part of the paper for the Altimetry implementation, the nodal modulations have been accounted, with an amplitude and phase slightly varying. The nodal variations have been implemented with the pytide module (https://github.com/CNES/pangeo-pytide). We propose to clarify this in the manuscript, thanks for this good point!
However, we did not note a sustancial effect with respect to no variations (we did the test on one tile), only a minor effect with less than a few percent differences in the performances.

*The English language usage is sometimes awkward or non-standard, especially with regard to capitalization. I am not evaluating it or going to list all the potential edits during this reading.*

Thanks for noting, we hope we fixed all capitalization issues, and tried our best to improve the English language in the new version.

*Smaller comments:*

*l14-15: Not sure where they get the 70% phase-modulated.*

Indeed this number does not appear in the 2017 Zaron paper. We propose to comment the Figure 9 from this same paper, suggesting that the fraction of unstationary internal tides exceeds 50% in Equatorial regions and Strong Western boundary currents. 70% for a global average was certainly an overestimation from a mis-reading of that figure.
This is clarified in the new manuscript.

*l35: "covariances" --> "spatial covariances"?*
Correct

*l49: specify, "$x_i$ and $x_j$ are uncorrelated for $i \ne j$"*
Yes, and we also mentioned "two components" instead of "N". This is also corrected.

*l62: Here it is specified that $x_1$ refers to the time-series of a scalar. Aha. But after line 70, it is clear that $x_2$ is a two-component vector containing the harmonic constants of the high-frequency component.*
Yes, and see our answer above : it was indeed unclear as we mentioned two components earlier.

*l75: The reference to localization should either be dropped or explained precisely what is meant.*

Yes indeed, we initially wanted to point out the necessity of inverting globally, as opposed to many situations where localization is implemented for practical reasons, But this is already discussed later. We agree it is more clear not to mention the non-localization here.

*p4, last line: How does it differ from harmonic analysis? If it is identical, then say so.*
Not identical. We propose the following explanation:
For the harmonic part, this separate estimation \added{performs nearly as a harmonic analysis, except that the finite $\mathbf{B}$ and non-zero $\mathbf{R}$ matrices tend to slightly reduce noise contamination with respect to harmonic fit.}

*l80: By "sequential estimation" do you mean that the low-frequency component is estimated by itself from the entire time series, and then this estimate is subtracted before estimating the high-frequency component? This usage of "sequential" is confusing since the term might also refer to sequential estimation (i.e., a Kalman filter) which sequentially processes the observations in time.*
Yes. Indeed, we did not realize the potential confusion. We could propose "alternate estimation" which is actually closer to what is done, since the sequence is done in alternate order for each component evaluation.

*l95-l100: This is very good discussion of bias in this context.*
Thanks!

*l125: Can you support your assumption that no correlation exists between the components (l116) by saying that the \Gamma_k are chosen to approximately diagonalize the state covariance? If you could provide some observational data to support the choice of \Gamma_k, that would be even better!*
*Aha: now I see the mention of this later, around l130.*

The different justifications were indeed incomplete and appeared in misleading order. We propose to re-write the whole paragraph presenting the Q matrix, directly mentioning the two main assumption : (1) the components are assumed orthogonal and (2) each given component is diagonalized (by construction). This is why the global Q matrix can be diagonal.

*l161: I believe the reference should be to Fig 4, not Fig 3? You will probably need a reference or short discussion to explain what is a "representer".*

Correct. We propose to add a description of the representor, but in the previous paragraph (regarding mesoscales) when it is introduced for the first time. In this paragraph, we also propose some clarifications, as the representer is shown for a given baroclinic mode (the first one) and for a given tidal component (M2). We believe the whole paragraph needed some clarifications, as proposed in the new version.

*Fig 4: Was the representer shown in the right panels constructed from the 12-equiangular basis elements? I am surprised that it is as radially-symmetric as shown.*

Yes, we believe that the number of directions must be related to the spatial extension of the plane waves. The longer it is, the higher the number of directions is needed to fulfill the decomposition. We propose a clarification of this point in the new manuscript.

*l173: This is a good compromise between domain size and degrees of freedom.*
Yes, we initially tried to inverse over 30 by 30 degree tiles, but the computation was too heavy...

*Fig 5: Did you subsample or average the observations in the along-track direction? Or did you use 1 Hz data? Why not show the same lat/lon window in each panel?*

Oh yes, we did, and actually a paragraph was clearly missing regarding the input dataset, as pointed out by another reviewer. As for the subsampling, we considered ⅓ Hz after averaging every three points. We verified that this averaging did not impact the results (sensitivity tests, not shown, suggested an impact beyond 4 consecutive points averaged). The new version of the manuscript has a new paragraph presenting the dataset.

*l231: How were the diagnonals of the Q matrices chosen initially? What information was used to estimate the variances of the signals?*

They were chosen according the the along-track Altimetry spectral database. Now that we propose a paragraph of clarification for the Q matrix, we believe this point l231 is also clarified.

*Fig 8 and discussion: Usually "signal variance" refers to the data, but I believe you are using it to refer to the variance of the estimated signal. Perhaps this could be clarified. My interpretation is based on the fact that you note the explained variance is larger than the "signal variance" in cases 2 and 3.*
Yes, we propose to use the following terminologies : "variance of the solution" and "explained variance after applying the solution as a correction to independent altimetry data". We hope now the figure and referring text is more clear.

*Maybe I missed it, but no where do I see discussion of what altimeter missions were used. It looks like CryoSat-2 is in the post-2017 validation dataset, but is this the only mission used?*

Yes, we realized that the validation dataset was not properly  described. New sentences have been added, and also refer to the new paragraph describing the main dataset used to compute the solution. The dataset refers to all satellite missions post September 2017.

*l235: Why are you using only a year for the validation period?*
It was a typo, it is actually 2 years. We performed the analysis in 2020 and at that time, our series ended in December 2019.

*Fig 10: Please state the units of the comparisons (cm^2, I think?).*

Yes, done!

*Why does the bias problem not seem to be as large as suggested by Fig 5? Perhaps I do not understand your sequential estimates, and they differ more significantly from the approach used in HRET. Or, maybe your low-frequency solution obtained here is quite different from the Duacs/Ssalto-based mesoscale correction used in HRET. Based on your Fig 5, I would have expected your estimate to explain a lot more variance than HRET.*

We think this point is now better supported by the new analysis, especially the Figure 4 showing that the MIOST solution is globally more energetic than HRET, especially in zones of high internal tides. So even id the net gain of variance reduction is overall moderate, the signal is stronger. In other words, we could say that MIOST resolves overall more signal, but also introduces slighly more errors.

*Table 1: What do the percentages refer to (is the decimal point placed correctly?)? Please label the sub-tables with M2 and K1.*
We propose a new table in the next version of the manuscript, more synthetic and with correct number of digits for percentage display.

---

## Author Comment (AC2)

**RC2: 'Comment on os-2021-80', Anonymous Referee #2, 20 Oct 2021**

We thank this anonymous reviewer for the relevant comments. We also propose a point-by-point response, sometimes referring to responses already developed in other reviewer questions above.

*This manuscript tackles an important question which is how to separate well the mesoscale variability from the internal tides in the altimetric record, with a focus on the internal tide component which is coherent. The approach is innovative, and the results are tested first on artificial fields, and then on the real altimetric sea level record, with a validation and estimation of the skill with recent data not used in estimating the solution (for the internal tides).*

*The pproach relies on a set of assumptions on the respective spectral characteristics of the meso-scale variability and the tidal characteristics. The tests are done assuming a certain spectral shape of the meso-scales and tides which follow the classical (linear) dispersion characteristics and are low order (1 or 2, depending on the tidal mode). The tests indicate that with these assumptions, the joint inversion approach (which is numerically rather heavy) performs better than separate approachs.*

*I wonder whether the authors could go further and estimate how much the gain depends on the spectral shape. After all, it originates from the overlay of the time-space spectra of the meso-scales and of the tides. One can also wonder how sensitive is it to the exact shape of spectrum. It could be interesting to test different shapes overlapping more or less.*

This is an interesting question, as the solution indeed relies on a number of assumptions that were necessary to extract specific signals among sparse observations. This is one of the difficulties : we need to implement enough assumptions to have a reduced basis with sufficiently

low degrees of freedom, but this reduced basis must still contain the signals we want to reconstruct, with correct prescription of variance.

We made different tests, in small domains to work with reasonable computing costs. We first verified that increasing the prescribed spectrum for mesoscale tends to overestimate the mesoscale solution, with some leakage from internal tides. And vice versa. We then verified that with a correct spectrum, and with a noise matrix set up accordingly with altimetry noise plus representativity errors, we could minimize the error with respect to independent Altimetry data. This is why we ended up using a database of spectra, varying regionally to treat the problem globally, with noise floor removed as in Dufau et al., 2016, from the AltiKa satellite.

The descriptions regarding how we use the spectra were poor in the first version of the manuscript, we now propose a more detailed version.

*What is the impact of the assumptions on spectral characteristics for the mesoscales, as well as for using a specified dispersion relationship with modeal decomposition, extending only to order 2 or 1 depending on the tidal component, of course compounded by the use of a (spatial) Hamming window. The width of this wndow has to have an impact. What fully motivates the choice?*

We answered the first part of the question above for mesoscales, and for the internal tides, we also verified that the solution was sensitive to the variance prescription for each mode of each tidal component. However, it is more difficult to have a direct observation of how much energy must be prescribed for coherent internal tides. Although we could use some estimations such as presented in Zaron 2019, based on the frequency width of internal tide peaks from Altimetry spectra, we specified the same 2cm**2 variance fore modes 1 and 1cm**2 for modes 2, which is more or less the maximal values in the zones of highly coherent and strong internal tides.  So this is certainly why in some region of low coherent internal tides, we tend to slightly overestimate internal tides, even though the consideration of mesoscales helps to mitigate the overestimation. This can be observed for instance on the Figure 4 of this document, in the Equatorial Pacific (where internal tides are known to be incoherent).

Then the question of the spatial Hamming window is also a very interesting one, and the impact of its size is crucial, which is certainly not discussed enough in the first version of the manuscript. The shorter this size is, the more possibilities (or degrees of freedom) there are, which is obviously more prone to leakage; Inversely, if we increase the length, we only restrain the solution to large-spatially-coherent plane waves, excluding the others.  Setting this length is therefore a complex balance, and we ended up choosing 2.5 times the spatial wavelength of the wave (as it can be visualized on Figure 3 of the manuscript, upper panels), which turned out to give good results w.r.t. Independent data. This factor is set the same everywhere and for all modes (1 and 2).

We propose to update the manuscript with such details.

*Section 3.1.1 summarizes the choices made in Ubelmann et al (2021) in a few sentences. This is fine not to present in details what is in this paper, but one is left a little bit wondering about what is been done. I was in particular wondering whether the choice to fit the covariance on the*

*altimetry mapping covariance, which filters out some of the smaller oceanic spatial scales has an impact on the internal tide solution.*

Good point, and this is why we tried to resolve (as much as we could) the mesoscales at the wavelength on the internal tides, say 200-120 km for the first mode and 100-60km for the second mode. To to so, we defined the smallest wavelength of mesoscales at 80km. This does not mean we do resolve mesoscales at such wavelength all the time, but that we consider them and partially represent them (near observations). This was indeed not very clear in the manuscript, as we commented that the covariance representor was similar to the one in Aviso mapping. We propose to modify it with the above explanation, and mention that our actual covariances do allow shorter scales.

*Also, when mentioning the full altimetric record, it should be indicated what is the data set. I assume that the adjustments between the different altimetric missions (and other corrections and filtering of the data along track, but that I am less sure) are performed before hand. Have these steps (if done) some implication on the internal tide characteristics that will be afterwards retrieved.*

Yes, this was definitely missing, as written above in the response to another reviewer comment, we have added a paragraph dedicated to the input dataset.

*My other comments are minor and could easily be fixed:*

Thanks, we do appreciate the following grammatical corrections that we fixed among others, we hope the new version will be improved.

*107: For each component k…*

Done

*115: index p should be explained.*

Yes, p was actually N (corrected) and N, the number of components, is now defined. Thanks!

*142 'only mode-1 is considered…'*

Yes

*155: '(sources and sinks)'*

Yes

*157: 'not too large'*

*Yes*

*162: I assume '(upper panel) …'*

Yes

*182, I don't understand the end of the sentence?*

We propose the following:

The matrix is therefore more dense (sparse in space only), as illustrated on the left panel of figure \ref{Gmatrix} : \added{the element contains more data (covering the 25-years of altimetry).}

*203: 'for each component'*

Yes

*207: I am not sure I got the end of the sentence: why 'supposedly'?*
*We propose to remove 'supposedly', indeed, this was because initially we prescribed the possibility of K1 waves even above the critical latitude 30°. In the version of the paper, the K1 waves are prescribed only in the -30°+30° band.*

210: 'the stationary persists' (word missing?)
Yes : stationary assumption

249: why is the Cryosat-2 mission specifically mentioned at this point (and not earlier)

This was a mistake! Initially we used Cryosat, but as now described in the paragraph above, we use all Altimetry data for validation, in the Spetember 2017 -December 2020 period. Thanks for noting!

257: 'could be an interesting next step…'
Yes, corrected.

---

## Author Comment (AC3)

**CC1: 'Comment on os-2021-80', Zhongxiang Zhao, 19 Oct 2021**

*This is a nice work that will draw interest from the altimetry community. It provides another new approach to map internal tides from satellite altimetry. It is a very cool method. It should be accepted with some minor revisions. In particular, the writing has a large room to improve.*

We thank Zhongxiang Zhao for providing community comments that are very useful in many aspects.

*L1, 'Ocean Surface Height', capital letters!?*
Thanks, we hope we fixed all problems regarding capitalization.

*L3, 'reduced-order basis with conjugate gradient resolution ' is vague in abstract, please specify.*
Yes, we propose to rephrase as follows: "The inversion is performed \deleted{through} \added{in reduced-order basis of topography and practically achieved} with conjugate gradient"

*L6, should be Msub2 etc.*
We apologize, we do not understand the suggested edit?

*L8, 'benefits' better be 'performance'?*
Yes, and we would propose this modified sentence :
From the solution, we use altimetry data after 2017 for an independent validation, to evaluate the performances of the simultaneous inversion and compare it with an existing model.

*L10, 'Internal Tides', capital letters?*
Corrected, thanks

*L11, '150 km ... and below' is not accurate, better say 'below 200 km '*
Good suggestion!

*l12, 'at first order'? Please explain or just cross out.*
Yes, so we would propose "affected by vertical stratification"

*L13, 'seasonal' a convincing paper is Zhao JPO (2021)*
Yes, the reference is added

*L17-18, 'O...G...C...M' no need in uppercase*
Ok

*L18, 'accurately' is not accurate*

Yes, we would propose 'with realistic amplitude'

*L21, 'later' or 'latter'?*
Yes, latter indeed

*L47, EQ2, move R definition here from a later place*
Yes, done

*L90, here and later, why do you repeat it '100' times? Any reason or criterion? Please give more details*
100 is to get sufficient realization for estimating the dispersion of errors on Figure 2 (especially right panel). So there is no rational criterion, 10 would have been too small to get smooth spectra on figure 2, 1000 would have worked, with would have given too much points to plot on the left panel. So 100 was a good compromise for this experiment that is purely qualitative. We propose some details in the new manuscript.

*L97, 'precisely'?*
Yes, we propose to delete this word.

*L143, Did you test with mode-2 S2 and O1?*
Yes, we did test, but the variance reduction tests against independent data suggested that the solution obtained for mode 2 of S2 and O1 contained more errors than signal, so we decided to consider only mode 1 for these secondary tidal components..

*L151, '3.5 m/s in the tropics'? I think higher speeds are at high latitudes, mainly due to the effect of f in EQ12.*
Yes, we mention here the first baroclinic phase speed in the Chelton et al. climatology. We agree that the induced wave speeds are higher at higher latitudes because of Eq. 2.

*L166, 'c2=c/2' is wrong! Please read Figures 3 and 4 in Zhao 2018 (JGR)*
Thanks for pointing this, c2 is indeed a more complex computation based on the density profile. This would actually bring some room for improvements of a future version of internal tide solutions. We propose to add a sentence explaining this is a crude approximation, that may be revisited in a next version.

*Figure 6, please give unit*
Yes, and we also forgot the variable name. All added now.

*Figure 8, based on the bottom panel, may we draw that MIOST-MSIT might underestimate internal tides, compared to MISOT-IT?*

Yes, we would rather say that MISOT-IT overestimates internal tides, because there is more contamination of the mesoscales in the internal tide estimation. Indeed, the green bars suggest than theMIOST-MSIT is better in term of variance reduction, even though the signal is weaker. So the MSIT solution remains better. Thanks, this was a good point, we propose a clarification in the text.

*Figure 9, for the bottom statistics, are you using the whole region above?*

Yes we do, it is true that the region extends beyond the Gulf-Stream, but as for the scores, they are clearly explained by the Gulf-Stream area where the signal is energetic. The contribution of the northern region is very small since there are almost no internal tides resolved. Since the goal of the figure is to make a qualitative point regarding the signal variance and the reduction of variance after applying the signal, we think this is fine to keep this (a bit large) domain.

*Table 1 is tedious: most of these numbers are very small. Please consider compress Table 1. These regions are defined in Carrere et al. (2021)? Many values are lower than 0.1%, may we say they are insignificant or within errorbar? 'Cryosat data' are from 2017-2018?*

*L262, I am not sure that 'opens the door for solving uncoherent internal tides'. There are TWO existing methods (Zaron; Zhao).*
We propose the following paragraph to explain how it could be extendented toward incoherent internal tides, of course with some caveats:
The methodology applied in this study \added{could also be extended to internal tides with phases varying seasonally, as already implemented in \cite{Zhao21}. Practically, we would introduce additional components, that could be built with the same plane wave basis, modulated with sines and cosines at 1-year frequency. Also, with the perspective of future wide-swath altimetry providing a larger volume of data, the uncoherent internal tides could also be tackled with time-finite modulation of the plane waves. This is obviously a challenge that will not be met everywhere, in particular where mesoscale fields dominate, but we hope that in some regions, some part of the uncoherent waves could be captured}.